# Long-Term Manure Amendment Sustains Black Soil Biodiversity by Mitigating Acidification Induced by Chemical N Fertilization

**DOI:** 10.3390/microorganisms11010064

**Published:** 2022-12-25

**Authors:** Lei Sun, Yongjie Yu, Evangelos Petropoulos, Xiaoyang Cui, Shuang Wang

**Affiliations:** 1Key Laboratory of Sustainable Forest Ecosystem Management-Ministry of Education, School of Forestry, Northeast Forestry University, Harbin 150040, China; 2Heilongjiang Academy of Black Soil Conservation & Utilization, Heilongjiang Academy of Agricultural Sciences, Harbin 150086, China; 3School of Applied Meteorology, Nanjing University of Information Science and Technology, Nanjing 210044, China; 4School of Engineering, Newcastle University, Newcastle upon Tyne NE1 7RU, UK

**Keywords:** long-term fertilization, black soil, microbial community assembly, sustainable agriculture, network analysis

## Abstract

The long-term use of chemical N fertilization may have a negative impact upon soil fertility and quality. On the contrary, organic fertilization is considered a sustainable development agricultural strategy. However, the remediation effect of organic fertilization on agroecosystems remains unclear. This study was conducted in a long-term (1979–2020) field experiment to investigate the influence of organic and chemical fertilizers on the soil microbiome assembly processes. The experiment consisted of six treatments: chemical N fertilization (N), double N fertilization (N2), organic fertilization (M), organic and N fertilization (MN), double organic and N fertilization (M2N2), and unfertilized control. The chemical N fertilization (N and N2) treatments significantly decreased soil microbial diversity, as well as soil pH, compared to the Control treatments (*p* < 0.05). MN and M2N2 treatments increased microbial diversity compared to that of N and N2 treatments. The combination of organic and chemical N fertilizer recovered the decreased microbial diversity to the level of the Control and M treatments, but the application of double organic fertilizer (M2N2) still showed a significantly lower microbial diversity than that of the Control and M treatments. From the results of the microbial community assembly processes, it was found that environmental filtering was induced by N fertilization, while organic fertilization developed a stochastic process and mitigated the role of environmental filtering in the community assembly process. An ecological network analysis showed that the decrease in *Acidobacteria* in organic fertilization treatments played a key role in mitigating the soil acidification induced by 40-year chemical N fertilization. It indicated that organic fertilizer could mitigate the decrease in soil fertility induced by chemical N fertilization. Higher environmental filtering effects in M2N2 than those in MN treatments suggested that the mitigation effect of organic fertilizer was weakened when double chemical N fertilization was applied in black soils. These results are helpful for a unified understanding of the ecological processes for microbial communities in the development of sustainable agriculture.

## 1. Introduction

Since the industrial revolution, chemical nitrogen (N) fertilization has been a significant agricultural strategy to promote crop yield [1]. Although there are clear benefits to this strategy, excessive chemical N fertilization can trigger severe environmental problems, such as soil acidification, soil salinization, and a decline in crop quality [1,2,3,4]. Organic fertilization is considered an alternative, more environmentally friendly strategy that can prevent the disadvantages related to the excessive use of chemical fertilization. It has been reported that organic fertilizers could lead to a lower degree of soil nutrition than that of chemical fertilizers [5,6]. Long-term fertilization field experiments in global regions have demonstrated that the simultaneous application of chemical and organic fertilizers is beneficial for modern farming; this is because it can promote crop yield while ensuring soil state and meeting the demand of sustainable and environmentally friendly agriculture [7,8].

The critical role of soil microbiota has driven numerous investigations into the relationships between the microbial community, soil nutrient cycling, and crop yield [9,10]. Despite the increasing number of studies focusing on the response of the soil microbial community and its corresponding phylogenetic diversity under different fertilization strategies, the response of microbial ecological assembly processes to long-term chemical and/or organic fertilizers still remains a substantial concern. Our previous study proved that chemical fertilization leads to stronger environmental filtering in the microbial community, compared to that from organic fertilization; meanwhile, organic fertilization provided more microbial niches and subsequently resulted in the higher stochastic process of the microbial community in the long-term fertilization of the southeast China region [11]. Hence, we hypothesize that the increased use of chemical fertilization could stimulate environmental filtering, while organic fertilization can drive the stochastic processes of the soil microbial community.

The black soil region in northeast China occupies approximately 0.2 million km^2^, with one fifth of that being used for agricultural purposes [12]. According to the statistical data of the Food and Agriculture Organization of the United Nations (FAO) and National Bureau of Statistics of China (NBSC), the region yields approximately 150 million tons of crop production annually and accounts for >40% of beans production in China. However, due to the excessive agricultural practices and the inherent properties of mollisols, soil erosion and acidification are phenomena that have become increasingly apparent in recent years. Xun et al. [13] compared the differences in the microbial community assembly processes in the soil suspension between black soil and red soil, using amplicon sequencing of the bacterial 16S rRNA gene. It has been demonstrated that pH strongly drives the community assembly, and that the keystone taxa appeared as *Acidobacteria*, *Actinobacteria*, and *Nitrospira* in the suspension of black soil. Meng et al. [14] reported that an increase in soil organic matter can improve the soil physicochemical properties and crop yield in the black soil region of China. Therefore, we hypothesize that the organic fertilization could shift the keystone taxa, shift the microbial community assembly processes, and subsequently influence the crop yield in black soil regions.

To verify our hypothesis, amplicon sequencing, targeting the soil bacterial community, was applied to a 40-year fertilization field experiment, established in October 1980 at Harbin, China. A null model and co-occurrence network analysis were calculated to reveal the different responses of the phylogenetic structure, the ecological assembly, and the exclusion links to chemical and organic long-term fertilizations. We also wanted to look for the key environmental factors and potential key microbial groups which drive community assembly processes when subjected to long-term fertilization in black soil regions.

## 2. Materials and Methods

### 2.1. Long-Term Field Experiment and Soil Sampling

Soil samples were collected from a long-term filed fertilization experiment at Harbin, Heilongjiang Province, China (45°40′ N, 126°35′ E). Since 1980, a wheat–maize–soybean rotation system has been constantly applied in this failed experiment. The study site is a typical monsoon climate, with a mean annual temperature of 3.5 °C, annual precipitation of 575 mm, and an annual evaporation of 1315 mm. The soil in the study site is a typical black soil. Each fertilization treatment was performed in a 9 m × 4 m field. In this study, six fertilization regimes were selected: no fertilization (Control), manure fertilization (M) (1.8 t horse manure ha^−1^ y^−1^), normal nitrogen fertilization (N) (150 kg N ha^−1^ y^−1^), manure and nitrogen fertilization (MN) (1.8 t manure plus150 kg N ha^−1^ y^−1^), double nitrogen fertilization (N2) (300 kg N ha^−1^ y^−1^), and double manure and double nitrogen fertilization (M2N2) (3.6 t manure plus 300 kg N ha^−1^ y^−1^). Each treatment was performed in triplicate with a field plot of 9 m × 4 m (length × width).

Soil samples were collected from the A horizon (0–20 cm) at ten points along a zigzag line using an auger boring. The subsamples were mixed after the removal of visible roots, stones, and soil fauna. A 100-mesh sieve was used to homogenize the samples prior to the chemical assay. The soil’s physiochemical properties were measured by the same method in our previous studies [11]. All tests were performed in triplicate. The soil samples for the molecular analysis were kept in a −80 °C freezer until use.

### 2.2. Molecular Methods

The soil genomic DNA was extracted from the same amount of soil (0.5 dry soil in equivalent) by using a FastDNA^®^ SPIN Kit for soil (MP Biomedicals, Santa Ana, CA, USA). The obtained soil DNA was dissolved in 50 μL TE buffer. The quality of DNA was performed by a spectrophotometer (NanoDrop ND-1000, Rockwood, TN, USA). Then, the qualified soil DNA sample was stored at −20 °C until further use. For the amplicon sequencing, soil bacterial fragments were amplified by the 515F and 806R primer set, according to the Earth Microbiome Project [15,16]. PCR reaction was carried out in 25-μL mixtures containing a 10.0 μL PCR master mix (TaKaRa, Dalian, China), 0.5 μL of forward and reverse primers (10 μM each), and 1 μL of template containing approximately 50 ng of genomic community DNA. After thirty-five cycles of 95 °C for 60 s, 50 °C for 60 s, and 72 °C for 105 s, a following extension step was performed at 72 °C for 10 min. The barcoded PCR products were normalized in equimolar amounts, then the homogenized DNA sample (mixed with 10% Phix control) was sequenced on a Miseq platform by using MiSeq Reagent Kit v3 (600-cycles-PE). Raw sequencing data were uploaded to NCBI Sequence Read Archive (SRA) database under accession numbers of SRR16217498 to SRR16217566.

### 2.3. Processing of the Sequencing Data

The analysis of the bacterial sequencing data was processed according to Divisive Amplicon Denoising Algorithm 2 (DADA2) pipeline [17]. Taxonomy was assigned to bacterial OTUs against a subset of the Silvar 138 database. We obtained a total of 3,123,843 bacterial 16S rRNA sequences, and between 30,158 and 59,199 sequences per sample. Sequences were randomly pruned to 30,000 per soil sample for downstream analysis. To reveal the changes in the phylogenetic structures between the different soil samples and to disentangle the underlying process influencing the community assembly in response to fertilizations, we calculated the mean pairwise phylogenetic distance (MPD) and nearest relative index (NRI); this was the degree of the non-random phylogenetic structure of communities [18,19]. The R package ‘picante’ was used to calculate the microbial community phylogenetic structure [20]. The microbial assembly indices, i.e., αMPD, αNRI and βNTI, were calculated according to our previous study [11]. In brief, αNRI was then calculated using ‘ses.mpd’ function and the observed αMPD was compared with the null distribution of αMPD generated by the 1000 randomizations of the ‘phylogeny.pool’ null model. The βNTI was calculated for phylogenetic community composition comparison purposes.

### 2.4. Statistical Analysis

Statistical analyses were carried out with the ‘vegan’ package in R platform. Differences at *p* < 0.05 were considered statistically significant. Molecular ecological network analyses were conducted to reveal the variations in the interactions between phylotypes responding to chemical and organic fertilizations, according to an online pipeline [21,22]. The correlation coefficient (ρ), with an absolute value over 0.90 and statistically significant (*p* < 0.01), was utilized for network analyses. The visualization of each network was performed with the interactive platform *Gephi* [23].

## 3. Results

### 3.1. Soil Microbial Diversity under Different Fertilizations

After more than 40 years of fertilization, the observed richness of the OTUs richness decreased approximately 10.3–13.6% with the normal chemical N fertilization treatments, compared to CK treatments (*p* < 0.05) (Figure 1a). When the organic manure fertilizer was applied to chemical N treatments, the observed richness of the OTUs r significantly increased (*p* < 0.05) (Figure 1a). The observed OTUs in MN treatments increased by approximately 10%, compared to N treatments. The double N fertilization treatments decreased the observed richness of the OTUs by 32.0–35.6%, compared to the CK treatments (*p* < 0.05). M2N2 treatments increased the observed richness of the OTUs by 30%, compared to N2 treatments. Correlation analysis showed that the observed richness of the OTUs was positively correlated with soil pH (*r* = 0.57, *p* < 0.001) and negatively correlated with soil available N (*r* = 0.50, *p* < 0.001) (Figure 1b).

### 3.2. Microbial Phylogenetic Diversity and the Inferred Ecology Process under Different Fertilizations

A principal co-ordinates analysis clearly showed that different fertilization methods clustered the microbial community (Figure 2). Specifically, the treatments lacking N fertilization were clustered at the lower left portion, the treatments with normal N fertilization (N and MN) were clustered at the upper left portion, and double N fertilization treatments (N2 and M2N2) were clustered at the right side of the plot. The additional application of organic fertilization (MN and M2N2) separated the microbial distribution from the chemical N fertilization treatments (N and N2) (Figure 2a). A null model analysis showed that long-term fertilization treatments significantly changed the microbial community assembly processes. The αMPD, which represented the phylogenetic relatedness between soil bacterial phylotypes, was significantly higher in N2 treatments, compared to that from other treatments (*p* < 0.05) (Figure 2b). N and M2N2 treatments had a lower αMPD than that of N2 treatments, but higher than that of MN treatments (*p* < 0.05). The control and M treatments had the lowest value of αMPD. To uncover the forces that shifted the community composition, the *α*NRI indices from different fertilization treatments were compared by using a null model (Figure 2c). We found that all *α*NRI values were significantly larger than zero (*p* < 0.001). The control and M treatments had significantly higher values than other treatments (*p* < 0.05), while MN and M2N2 generated intermediate values. This indicated that the co-occurring bacteria taxa in all treatments were more phylogenetically clustered than what would be expected if this was all driven by chance. To uncover the key environmental factors influencing microbial community assembly processes under different fertilization treatments, we conducted correlation analyses between βNTI and soil physic-chemical properties (Appendix A and Appendix A). Soil pH significantly decreased in chemical N fertilization treatments (N and N2), compared to M and Control treatments (Appendix A) (*p* < 0.05). A significant positive relationship was observed between βNTI and changes in pH (Figure 2b).

### 3.3. Response of the Bacterial Phylotype Co-Occurrence to Different Fertilization Regimes

A non-random co-occurrence network analysis was calculated to reveal the variations in the interactions between bacterial phylotypes, nitrogen and manure fertilizations in the black soils of northeast China (Figure 3). The modularity index value from each treatment ranged from 0.97 to 0.98, which was higher than 0.4, indicating that all of the co-occurrence networks were modularly structured networks. High N fertilization treatments (N2 and M2N2) achieved a higher number of nodes and edges, compared to normal N fertilization treatments (N and MN). The application of manure fertilization (MN and M2N2) decreased the number of nodes and edges, compared to the chemical fertilization treatments (N and N2). The average clustering coefficient (degree of how nodes are embedded in their neighborhood) and average degree (nodes clustering degree) showed a higher value in double N treatments (N2 and M2N2) than normal N treatments (N and MN). This indicated that the microbial community tended to more closely cluster together when the double N fertilization was applied. The dominant bacterial phyla in the co-occurrence networks were *Proteobacteria*, *Actinobacteria*, *Acidobacteria*, *Bacteroidetes* and *Chloroflexi* etc., and percentage of these phyla differed among different fertilization treatments. Specifically, the co-occurrence network analyses showed that the percentage of *Proteobacteria* and *Acidobacteriota* phyla in MN and M2N2 treatments decreased, compared to N and N2 treatments.

### 3.4. The Responses of Microbial Phylotypes to Soil Properties under Different Fertilizations

A phylogenetic tree analysis was conducted to reveal the dominant microbial phylotypes across all treatments in this study (Figure 4). It was found that *Acidobateria*, *Actinbacteria*, *Alphaprotebacteria*, *Gammapretobacteria* etc. were the predominant phyla across all treatments (Figure 4a). The distribution of microbial phyoltypes was significantly correlated with soil pH and the available N (*p* < 0.05). The percentage of *Acidobateria* was higher in N and N2 treatments, compared to that from MN and M2N2 treatments (Figure 4b). A positive correlation was observed between soil pH and the percentage of *Acidobacteria* (Figure 4c). The βNTI was found positively correlated with the changes in crop yield (Figure 4d).

## 4. Discussion

### 4.1. Organic Fertilization Mitigated the Decline of the Soil Bacterial α-Diversity Caused by Chemical N Fertilization in Black Soils

Long-term chemical N fertilization can have a negative impact on agroecosystems, such as soil acidification [24,25] and salinization [26,27,28]. So far, there are various arguments regarding the role of N fertilization in soil microbial diversity and how this can drive the issues mentioned above. For example, it was found that the abundance of microbial phylotypes was not significantly altered even after 30 years of chemical fertilization [11]; meanwhile, the α-diversity of soil bacteria significantly decreased in the fluvo-aquic soils. In this study, after a 40-year field fertilization in the black soil region, the soil microbial α-diversity significantly decreased >10% with N treatments and decreased >30% with N2 treatments, compared to the Control treatments. The main reason for the negative response in the microbial diversity might be due to the decline in soil pH for the N treatments. Previous studies have demonstrated that soil pH is a potential predictor of bacterial communities and their corresponding diversity in various ecosystems [29,30,31]. In this study, soil pH significantly decreased with the increase in chemical N fertilization. According to the theory of proton budgets, the stimulation effect of chemical N fertilization on the leaching of nitrates can drive the decline in soil pH in agroecosystems on a global scale [32]. Therefore, it is reasonable that soil microbial diversity decreases when chemical N fertilizer increases in black soils.

Organic fertilization is becoming a recommended strategy in agricultural sustainable development [7,9], while the mechanism underlying the potential process of organic fertilization remains relatively unknown. In this study, the single application of organic fertilization (M) did not significantly change the soil microbial diversity, compared to the Control treatments, even after 40 years of fertilization. This phenomenon corroborated the role of organic fertilization in sustainable agriculture to some extent. Comparing the difference in the microbial diversity between N and MN treatments, we found that the combination of organic fertilization and chemical N could increase the abundance of observed OTUs to the level of microbial diversity for the Control and M treatments. There were two likely causes for the mitigation effect of organic fertilization in the decrease in microbial diversity by N fertilization. One plausible explanation for this phenomenon is that the porous structure of the compounds in organic fertilization can slow down the water flow in soils [12]; thus, organic fertilizers can mitigate the erosion of soil by precipitation [33]. Subsequently, this provides more ecological niches for the microbes in the studied black soil. Another plausible explanation is that the carbon-based macromolecules in the organic fertilizer may absorb the anion (such as the nitrate introduced by N fertilization) [34,35]; this, then, increases the soil pH, which significantly positively correlates with the microbial diversity in this study (*p* < 0.05). We also found mitigation effects when comparing the microbial diversity between N2 and M2N2 treatments. However, interestingly, the application of organic fertilization could not amend the microbial diversity of that in the Control and M treatments, even when double organic fertilizer was applied (M2N2). This phenomenon supported that the application of N fertilizer should take place with more caution, since the excessive use of chemical N fertilization made the amend process more challenging from the perspective of the soil microbial diversity.

### 4.2. Organic Fertilization Mitigated the Role of Environmental Filtering on Microbial Community Assembly in the Black Soils

The phylogenetic signals showed a strong environmental filtering process in the studied black soils, which shifted the microbial community in the N and N2 treatments. This increase in the environmental filtering process could be attributed to the selective force of the nutrient inputs and the crop demand, since black soil is typically abundant in organic C and low in the ratio of C and N [11]. We then found that the application of organic fertilizer decreases the environmental filtering process and increases the stochastic or neutral process when comparing the differences in the microbial community assembly processes between organic fertilization (MN and M2N2) and chemical N treatments (N and N2). The ecological niches of partitioning are essential to the functional traits within a given ecosystem [10,18,26]. In this study, a network topology was performed to investigate the patterns of the microbial relationships under different long-term fertilization treatments. It was found that *Proteobacteria* and *Acidobacteria* occupied more ecological niches in the N and N2 treatments compared to MN and M2N2 treatments, while more microbial phylotypes in the ecological networks were also observed in the MN and M2N2 treatments. A possible explanation for this is that soil salinization and leaching processes were aggravated due to long-term chemical N fertilization and the decreased soil pH in N and N2 treatments [3,12]. Subsequently, *Proteobacteria* and *Acidobacteria* were stimulated in N and N2 treatments, as they can cope with low pH conditions. When the organic fertilization was applied, the conditions for soil microbes were improved; specifically, soil pH increased in MN and M2N2 treatments. Consequently, a more lenient environment stimulated the neutral/stochastic ecological assembly processes and more probable ecological functional traits in the MN and M2N2 treatment. These results are consistent with our initial hypothesis, propounding that the increased use of chemical fertilization could stimulate environmental filtering, while organic fertilization can drive the stochastic process of the soil microbial community. The ecological network analysis showed that chemical N fertilization stimulated *Acidobateria* and *Proteobacteria,* while organic fertilization decreased the link between the two phyla in the topology of the network. These results were consistent with those from a previous study on the distribution of bacterial phyla in agro-ecosystems across a global scale [36]. Statistical analysis results showed a positive relationship between soil pH and phylogenetic signals (*p* < 0.05). Soil pH was also found to be positively correlated with the percentage of *Acidobateria*. Thus, it can be inferred that the organic fertilization mitigated the increase in soil *Acidobateria* due to N fertilization, then it subsequently decreased the environmental filtering effects induced by chemical N fertilization. The significantly increased crop yield also supported the existence of more abundant functional traits in MN and M2N2 treatment, compared to N and N2 treatments. Interestingly, we found a higher environmental filtering effect for the M2N2 treatments than the MN treatments. This phenomenon suggests that the excessive use of chemical N fertilizer will likely get challenged during the remediation of soils with organic fertilizer, even when/if a double dose is used.

## 5. Conclusions

This study focused on the influence of organic fertilization on soil qualities, compared to the chemical N fertilization. This was from the perspective of the microbial community assembly mechanism and ecological networks. Our results indicated that soil acidification, as well as environmental filtering effects, were aggravated by long-term chemical N fertilization, while organic fertilization can mitigate these negative effects induced by chemical N fertilization. The mitigation effects of organic fertilization on soil qualities and biodiversity were significantly reduced when double chemical N fertilization was applied; even double organic fertilizer was amended in the 40-year field experiment. Ecological network analyses showed that the distribution of *Acidobacteria* and *Proteobacteria* significantly decreased in MN and M2N2 treatments, compared to N and N2 treatments, indicating that organic fertilization can mitigate the competition of these bacterial taxa induced by long-term N fertilization. A combination of null models and ecological network analyses can provide an overall understanding of the mechanisms in the development of a more sustainable ‘organic’ agriculture.

## Figures and Tables

**Figure 1 microorganisms-11-00064-f001:**
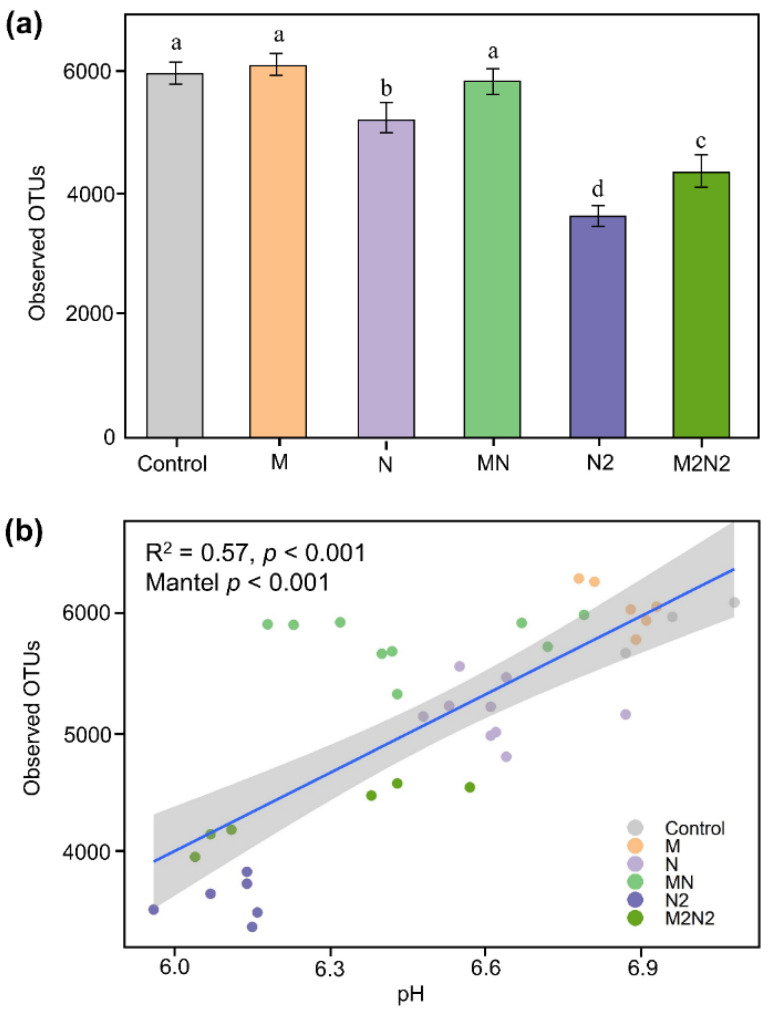
The impact of long-term fertilization to soil microbial diversity (**a**) and the relationship between observed OTUs richness and soil pH (**b**). Bars with different letters (shown in above) are significantly different (*p* < 0.05).

**Figure 2 microorganisms-11-00064-f002:**
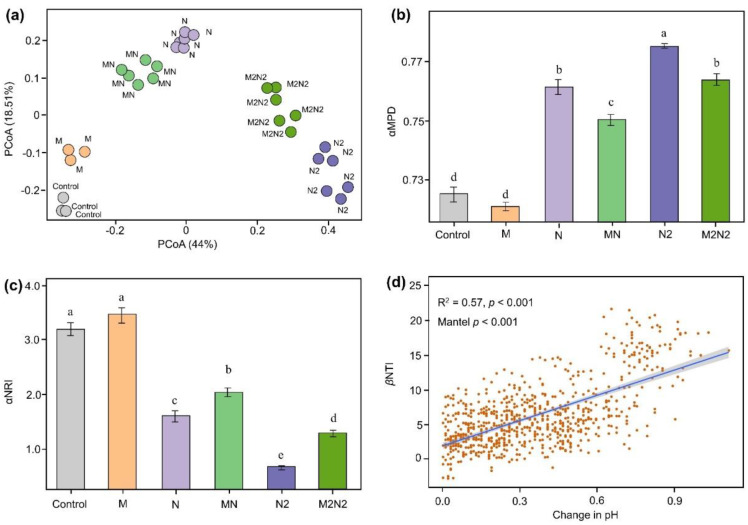
Phylogenetic signals under different fertilizations. (**a**) Principal co-ordinates analysis plot basing on Bray–Curtis distance; (**b**) Mean pairwise phylogenetic distance of α diversity (αMPD); (**c**) Nearest relative index of α diversity (αNRI); (**d**) correlation analysis between βNTI and changes in pH. Bars with different letters (shown in above) are significantly different (*p* < 0.05).

**Figure 3 microorganisms-11-00064-f003:**
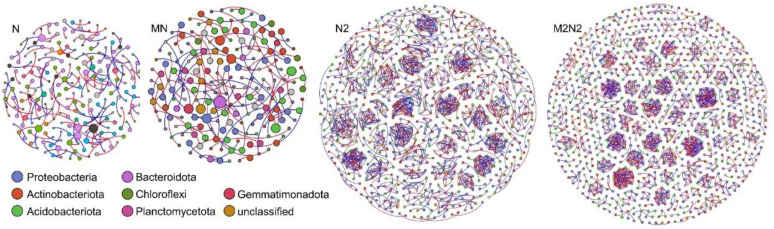
Network co-occurrence analysis of the bacterial communities within each fertilization treatment. A connection stands for a strong (Spearman’s R > 0.91) and significant (*p*-value < 0.001) correlation. The size of each node is proportional to the number of connections (that is, degree).

**Figure 4 microorganisms-11-00064-f004:**
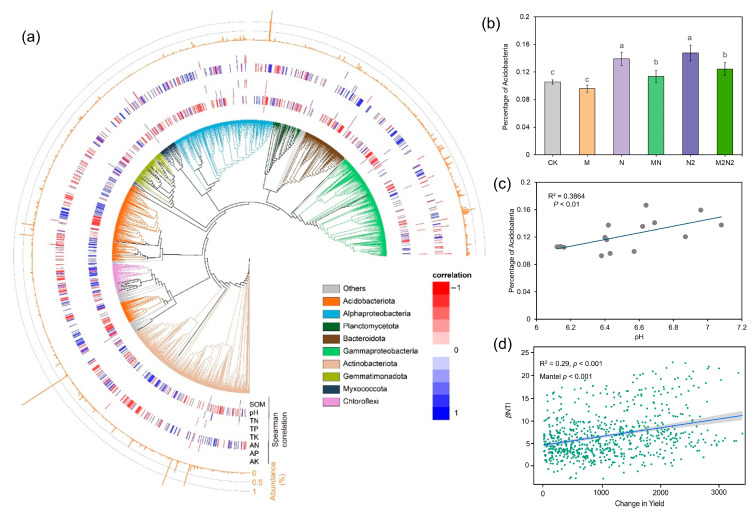
Phylogenetic tree of the microbial phylotypes and relationship between microbial community and soil properties. (**a**) Predominant phyla across all treatments; (**b**) Percentage of *Acidobateria* phylum in different treatments; (**c**) Correlation between the percentage of *Acidobateria* phylum and soil pH; (**d**) Correlation between βnti and crop yield. Bars with different letters (shown in above) are significantly different (*p* < 0.05).

## Data Availability

The raw data supporting the conclusions of this article will be made available by the authors, without undue reservation.

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
