# Peer review of "Long-Term Manure Amendment Sustains Black Soil Biodiversity by Mitigating Acidification Induced by Chemical N Fertilization"

_microorganisms, 2022, doi:10.3390/microorganisms11010064_

Round 1
Reviewer 1 Report
Fertilization strategy is associated with sustainability of agricultural. This manuscript performed an interesting study that linked the different fertilization treatments and soil microbiome, with the appropriate contrasting plots and numbers of replication, such studies are scarce and warrants merit. Though the soil bacterial community cannot reflect all the soil microbiome in agroecosystems, the results in this study can clearly reveal the significant differences of microbial communities between different fertilization treatments. The further statistical analysis in this manuscript demonstrated the key role of soil microbial community in the long-term organic fertilization treatments. The data and interpretation in this study is reasonable and reliable. It is a sound and solid study, which could give a heuristic and pragmatic value for the sustainable agriculture in the future. In my opinion, it can be accepted for publication after minor modifications.
Minor concerns:
1. Abstract section, Line 29-31: rephrase the sentence 'There are no significant differences… that from N2 treatments'.
2. Line 131-133: The sequencing procedure on Miseq platform should be described with more details, such as the information of Phix control.
3. Line 155, 'Differences at P<0.05 were considered statistically significant. ' Lines 163-165, ' After more than 40-year fertilization. compared to CK treatments (p < 0.05).’ The expression of P value should be unified in this manuscript.
4. Lines 193-199: The statistical analysis was not conducted between different fertilization treatments, although the data was put in the supplementary material. The significant differences of soil physic-chemical properties between different fertilization treatments is beneficial for the understanding of the background condition after the long-term fertilization, especially for the influence of chemical N fertilization on soil pH. As the title of this manuscript emphasized different status of soil acidification between chemical N fertilization and organic fertilization, the change of soil pH should be clearly described in this section.
5. Line 318: It seems Conclusion section is absent in this manuscript.
6. Reference section: The format of references is not unified, please carefully check the reference list and modify according to the Author guidelines on the website of MDPI.
Author Response
- Abstract section, Line 29-31: rephrase the sentence 'There are no significant differences… that from N2 treatments'.
Response 1: Thank you for the direct suggestion. The confusing sentence is rewritten. Please see Line 29-32 in the modified manuscript.
- Line 131-133: The sequencing procedure on Miseq platform should be described with more details, such as the information of Phix control.
Response 2: 10% Phix control was mixed to the DNA sample before sequencing on the Miseq platform. This information was added. Please see Line 135.
- Line 155, 'Differences at P<0.05 were considered statistically significant. ' Lines 163-165, ' After more than 40-year fertilization. compared to CK treatments (p < 0.05).’ The expression of P value should be unified in this manuscript.
Response 3: Sorry for the careless. The expression of P value is modified to a unified form in the modified manuscript. Please see Line 168.
- Lines 193-199: The statistical analysis was not conducted between different fertilization treatments, although the data was put in the supplementary material. The significant differences of soil physic-chemical properties between different fertilization treatments is beneficial for the understanding of the background condition after the long-term fertilization, especially for the influence of chemical N fertilization on soil pH. As the title of this manuscript emphasized different status of soil acidification between chemical N fertilization and organic fertilization, the change of soil pH should be clearly described in this section.
Response 4: The influence of chemical N fertilization on soil pH was described in the modified manuscript. Please see Line 201-203.
- Line 318: It seems Conclusion section is absent in this manuscript.
Response 5: The conclusion section has been added in the modified version. Please see Line 323-337.
- Reference section: The format of references is not unified, please carefully check the reference list and modify according to the Author guidelines on the website of MDPI.
Response 6: We have modified the references according to the MDPI format.
Reviewer 2 Report
The article does not bring completely new information. The results are expected and fit in with current knowledge. It confirms already known information on the effect of manure and mineral fertilisers on soil properties. Nevertheless, they are very useful because the data come from long-term experiments and their predictive value is great.
In terms of methodology, I'm missing information about the experiment. What quantities of fertiliser were applied to the plots? Are these typical (common) fertilizer rates in the area? What manure was used? What was the dose? Was it manure or slurry?
Figure 3 is a mystery to me. It is not mentioned anywhere in the text.
The difference between N2 and M2N2 (Fig. 3) is not clear to me from the graph. Can you describe it more?
The work is very nice otherwise.
Author Response
Response to Reviewer 2 Comments
Point 1: In terms of methodology, I'm missing information about the experiment. What quantities of fertiliser were applied to the plots? Are these typical (common) fertilizer rates in the area? What manure was used? What was the dose? Was it manure or slurry?
Response Point 1: Sorry for the missing information about the fertilization details. It has been added to the modified version. Please see Lines 110-114.
Point 2: Figure 3 is a mystery to me. It is not mentioned anywhere in the text.
Response Point 2: Sorry for the careless. The section 3.3 in the manuscript is the result of Figure 3, and we discussed these results in section 4.2. We forgot to add the quotation of Fig. 3 in the manuscript. It has been added to Line 208.
Point 3: The difference between N2 and M2N2 (Fig. 3) is not clear to me from the graph. Can you describe it more?
Response Point 3: Thanks for the direct suggestion. The detailed description of the differences between N2 and M2N2 in the network analyses is added to Line 221-223.
Reviewer 3 Report
Dear authors,
I appreciate the idea of your research, to analyze long-term effects of agronomic inputs on soil biodiversity. In the current context, such results are important to make future steps toward sustainability.
There are some suggestions that I consider will improve your manuscript.
Keywords - add one or two more keywords related to your work.
The introduction is clear, and only the last paragraph should be modified. Try to present your objectives/hypotheses in a more clear manner. This will improve the manuscript readability. This sentence should be moved to the aim and objectives paragraph: "Hence, we hereby hypothesize that the organic fertilization could shift the keystone taxa, microbial community assembly processes and subsequently influence the crop yield in black soil regions."
All the sections - Mat and Meth, Results and Discussion - are well written, present the results and compare them with other researches. I suggest you to include references in the 4.2. section of the discussion.
Overall, the entire manuscript is interesting and important for the current context.
Author Response
Response to Reviewer 3 Comments
Point 1: Keywords - add one or two more keywords related to your work.
Response 1: Thanks for the direct suggestion. We have added two more keywords. Please see Lines 46-47.
Point 2: The introduction is clear, and only the last paragraph should be modified. Try to present your objectives/hypotheses in a more clear manner. This will improve the manuscript readability. This sentence should be moved to the aim and objectives paragraph: "Hence, we hereby hypothesize that the organic fertilization could shift the keystone taxa, microbial community assembly processes and subsequently influence the crop yield in black soil regions."
Response 2: Thanks for your comments about the structure of the paragraph. It is an insightful opinion. In the introduction section we tried to propose two hypotheses. One hypothesis is that the increased use of chemical fertilization could stimulate environmental filtering. Another hypothesis is that the organic fertilization could shift the keystone taxa. These two hypotheses were proposed according to the statements in the second and third paragraphs in the introduction section. We still appreciate your insightful suggestion about the readability of our manuscript.
Point 3: All the sections - Mat and Meth, Results and Discussion - are well written, present the results and compare them with other researches. I suggest you to include references in the 4.2. section of the discussion.
Response 3: Thanks for the affirmation to our study. More references have been added to Section 4.2. Please see Lines 287, 292, 299 and 312.